# Interaction of Graphene Oxide Nanoparticles with Human Mesenchymal Stem Cells Visualized in the Cell-IQ System

**DOI:** 10.3390/molecules28104148

**Published:** 2023-05-17

**Authors:** Sergey Lazarev, Sofya Uzhviyuk, Mikhail Rayev, Valeria Timganova, Maria Bochkova, Olga Khaziakhmatova, Vladimir Malashchenko, Larisa Litvinova, Svetlana Zamorina

**Affiliations:** 1Institute of Ecology and Genetics of Microorganisms, Ural Branch of the Russian Academy of Sciences-Branch of Perm Federal Research Center, 614081 Perm, Russia; kochurova.sofja@yandex.ru (S.U.); mbr_59@mail.ru (M.R.); timganovavp@gmail.com (V.T.); krasnykh-m@mail.ru (M.B.); mantissa7@mail.ru (S.Z.); 2Department of Microbiology and Immunology, Faculty of Biology, Perm State University, 614990 Perm, Russia; 3Department of Microbiology and Immunology, Faculty of Biology, Immanuel Kant Baltic Federal University, 236041 Kaliningrad, Russia; hazik36@mail.ru (O.K.); vlmalashchenko@kantiana.ru (V.M.); larisalitvinova@yandex.ru (L.L.)

**Keywords:** graphene oxide nanoparticles, human mesenchymal stem cells, PEG, cell viability, cell growth, Cell-IQ, flow cytometry

## Abstract

Graphene oxide is a promising nanomaterial with many potential applications. However, before it can be widely used in areas such as drug delivery and medical diagnostics, its influence on various cell populations in the human body must be studied to ensure its safety. We investigated the interaction of graphene oxide (GO) nanoparticles with human mesenchymal stem cells (hMSCs) in the Cell-IQ system, evaluating cell viability, mobility, and growth rate. GO nanoparticles of different sizes coated with linear or branched polyethylene glycol (P or bP, respectively) were used at concentrations of 5 and 25 μg/mL. Designations were the following: P-GOs (Ø 184 ± 73 nm), bP-GOs (Ø 287 ± 52 nm), P-GOb (Ø 569 ± 14 nm), and bP-GOb (Ø 1376 ± 48 nm). After incubating the cells with all types of nanoparticles for 24 h, the internalization of the nanoparticles by the cells was observed. We found that all GO nanoparticles used in this study exerted a cytotoxic effect on hMSCs when used at a high concentration (25 μg/mL), whereas at a low concentration (5 μg/mL) a cytotoxic effect was observed only for bP-GOb particles. We also found that P-GOs particles decreased cell mobility at a concentration of 25 μg/mL, whereas bP-GOb particles increased it. Larger particles (P-GOb and bP-GOb) increased the rate of movement of hMSCs regardless of concentration. There were no statistically significant differences in the growth rate of cells compared with the control group.

## 1. Introduction

Graphene is a gapless semiconductor that is currently being actively used in microelectronics and materials science [1,2,3] for various applications including the development of narrow-band electromagnetic absorbers [4] and metamaterials [5]. Due to the complexity of scalable production, its functional derivatives are more suitable for some applications. For example, graphene oxide (GO) offers a favorable alternative for use in biomedicine and optoelectronics due to its ease of fabrication, water solubility, and optical properties [6,7,8,9]. The graphene surface in GO is derivatized with epoxy, hydroxyl, and carboxyl groups, which enable it to form hydrogen bond-stabilized water suspensions [10,11,12]. In addition, GO has a large surface area available for functionalization and excellent mechanical properties [13,14], which make it attractive overall for electronics (LEDs and solar cells), tissue engineering, and drug delivery [15,16,17,18]. GO can be used as a basis for nanoscale sensors that detect small molecules such as NO_2_ [19], proteins [20], influenza virus strains [21], and DNA strains [22]. It can also be used in pH measurement based on fluorescence to detect the microenvironment of tumors [23]. GO is effectively internalized by cells and shows stable fluorescence emission within cells as well as low cytotoxicity at the concentrations used for visualization [24,25]. This makes GO a potential candidate for drug delivery and in vitro or ex vivo visualization that can be used to detect and treat cancer [26]. However, in order for such applications of GO to become widely adopted, a solid understanding of the distribution of GO nanoparticles in vivo is required as well as knowledge of how it might interact with cells of different tissues.

Human mesenchymal stem cells (hMSCs), also known as multipotent mesenchymal stromal cells [27], are undifferentiated cells with the ability to self-renew and differentiate into various mesenchymal tissues, mainly bone, cartilage, and adipose tissue. According to the International Society for Cell Therapy, cells that possess the following characteristics can be considered human multipotent mesenchymal stromal cells [27]:-Adhere to plastic;-Differentiate in vitro into osteoblasts, adipocytes, and chondroblasts;-Express CD105, CD90, CD73, CD44, and HLA-DR;-Lack CD45, CD34, CD14, CD11b, CD79, and CD19.

Mesenchymal stem cells were first isolated from bone marrow in 1968. Since then, they have been isolated from many other tissues, including adipose tissue [28], perivascular networks [29], dental pulp [30], muscle, dermis, and embryonic tissue [31]. MSCs derived from bone marrow and adipose tissue are widely used in regenerative medicine [32]. It has been reported that drug-loaded hMSCs can deliver therapeutic cytokines to sites of injury or inflammation, making them a potentially useful tool in regenerative medicine and anti-tumor therapy [33]. Thus, hMSCs are attractive candidates for use as carriers of therapeutic agents and bioactive materials to specific target sites.

Cell-IQ^®^ (CM -Technologies, Tampere, Finland) is a fully automated system for continuous in vitro cell imaging [34]. It includes an inverted phase contrast microscope with a built-in video camera and a climate chamber that maintains a constant temperature (37 °C) and CO_2_ concentration (5%). Inside the climate chamber is a movable stand for a cell culture plate. Every 30 min, the system takes photographs in predetermined areas of the plate. The integrated software allows tracking of individual cells, including changes in their morphology and movement. The aim of this study was to investigate the interaction between GO nanoparticles and hMSCs with the Cell-IQ^®^ system.

## 2. Results

### 2.1. Determination of the Relative Numbers of Live and Dead Cells after 24-h Incubation with GO Nanoparticles

Data on the number of live, apoptotic, and dead cells after 24-h incubation with GO nanoparticles, as determined by flow cytometry analysis, are shown in Figure 1. Cultivation of human mesenchymal stem cells in the presence of graphene oxide nanoparticles of types P-GOs, bP-GOs, and P-GOb at a concentration of 5 μg/mL did not result in a statistically significant decrease in the number of viable cells compared with the control group. The number of viable cells in these groups, as in the control group, was approximately 90%, which corresponds to the viability of the cells before the start of the experiment. At the same time, incubation of cells with these types of nanoparticles at a higher concentration of 25 μg/mL significantly decreased the relative number of viable cells. A decrease in the number of living cells correlated with an increase in the number of both dead and apoptotic cells, with a greater increase in the number of dead cells in all cases. The ratio between the number of dead cells and the number of apoptotic cells (D/A) in these samples was 1.74 ± 0.17, whereas in the control group D/A = 1.1.

The greatest cytotoxicity among the samples studied was exerted by the bP-GOb nanoparticles. In this sample, cell viability decreased significantly after 24 h of cultivation, regardless of the concentration of nanoparticles used. Interestingly, the low concentration of bP-GOb resulted mainly in an increase in the number of dead cells (D/A = 2.54), whereas at a high concentration the loss of cell viability was mainly due to an increase in the number of apoptotic cells (D/A = 0.39).

Thus, it was shown that all particles studied had a cytotoxic effect on human mesenchymal stem cells at high concentrations. At low concentrations, the cytotoxic effect was observed only for bP-GOb particles. In all cases, there is a decrease in viability, mainly due to an increase in the number of dead cells. A decrease in viability due to the predominant increase in the number of apoptotic cells was observed only for the bP-GOb sample at high concentration.

### 2.2. Evaluation of Particle Internalization by Cells

After a 24-h incubation of cells with all samples of GO nanoparticles (except P-GOs at high concentration and bP-GOs at low concentration), a statistically significant increase in the number of highly granular cells was observed compared with the control group (Figure 2). This fact might indicate that either adhesion of the particles to the cell surface or their internalization occurs. We did not find any correlation between the concentration of particles and the percentage of high-granular cells, which may be due to several reasons. For example, particles that were weakly adsorbed on the cell surface might be lost during washing before flow cytometric analysis. In addition, we cannot exclude the influence of the physiological state of the cells (transition to apoptosis, death) on the granularity.

### 2.3. Evaluation of Cell Growth and Migration Activity in the Cell-IQ System

A total of 188 images were acquired for each well of the cell culture plate (47 images for each area of visualization) (Figure 3). Adhesion of particles on the cells is clearly visible in these images (Appendix A). This is particularly noticeable in wells with high concentrations of samples. In addition, 24 h after the start of cultivation, areas without nanoparticles can be seen on the images. These correspond to the trajectories of hMSC movement as the cells have adhered/internalized most of the particles they encountered along the way.

To assess the change in the number of cells in the visualization areas, the cells in every other image were counted. A pivot table was created based on the data obtained (Appendix A). There were no statistically significant differences in either the increase in cell number after one day of cultivation or in the value of the K1 coefficient compared with the control group (Figure 4). In the evaluation of cell activity (K2), a statistically significant decrease in the activity of the cells cultured with the P-GOs sample at high concentration was observed compared to the control group. The opposite was true for the bP-GOb sample. At the minimum concentration, it apparently enhanced cell activity, resulting in a statistically significant increase in K2 levels compared to the control group. No statistically significant differences were found in the other samples compared to the control group. In general, it appears that low concentrations of GO nanoparticles contributed to an increase in cell activity, while high concentrations, on the contrary, had an inhibitory effect.

In addition, the velocity of cell movement was evaluated (Figure 5). An increase in the speed of cell movement was observed for all samples and concentrations studied compared to the control group, while this was statistically significant only for the P-GOb and bP-GOb samples.

To assess the effect of the samples on the functional properties of the hMSCs, the change in cell activity (K2) was evaluated. It can be seen that the concentration of the nanoparticles has a significant effect on the K2 value. The average K2 value for areas of visualization containing samples at a concentration of 25 μg/mL is lower than that of the control group, whereas it is higher for areas containing samples at a concentration of 5 μg/mL. Visually, it was observed that isolated cells moved farther and faster, while they slowed down when they were near other cells.

Overall, it was found that a high concentration of P-GOs particles statistically significantly reduced the value of the K2 coefficient, while high concentrations of bP-GOs particles increased K2. A statistically significant effect of other particle types on the K2 coefficient was not observed, but an upward trend for the K2 value can be seen for the samples in the following series: P-GOs → bP-GOs → P-GOb → bP-GOb (Figure 4). Moreover, a statistically significant increase in the speed of cell movement was observed for P-GOb and bP-GOb type particles at both concentrations, while no change in cell speed was registered for P-GOs and bP-GOs particles compared to the control.

## 3. Discussion

For all samples studied, the adhesion/internalization of GO nanoparticles by the cells was visually observed in the images captured by the Cell-IQ system. This observation is confirmed by the number of highly granular cells determined by flow cytometry.

GO nanoparticles of the P-GOs, bP-GOs, and P-GOb types at a concentration of 5 μg/mL show a low level of cytotoxicity during short-term incubation with human mesenchymal cells, as there are no significant differences in the relative numbers of live and dead cells between the experimental and control groups. A high concentration of graphene oxide nanoparticles during short-term incubation results in a significant decrease in the number of live cells compared to the control group for all types of nanoparticles used in this study. Graphene oxide nanoparticles of the bP-GOb type have the most significant negative effect on the viability of human mesenchymal stem cells during short-term cultivation by significantly reducing the relative number of living cells, regardless of the concentration used.

Multiple mechanisms of cytotoxicity of GO nanoparticles have been described by other authors. The main mechanisms include interaction of cells with extremely sharp graphene edges [35], generation of reactive oxygen species (ROS) [36], and trapping of cells within aggregates of GO nanosheets [37]. In this study the mechanisms behind cytotoxicity of GO nanoparticles were not investigated and we possess no data regarding ROS generation or levels of oxidative stress the cells experienced. However, it is certain that the coating of GO nanoparticles by linear and especially branched PEG helped reduce the damage cells sustained from contact with sharp edges of graphene. Moreover, aggregation of nanoparticles on the cell surface was clearly seen by phase-contrast microscopy and may have contributed to cytotoxicity.

The nanoparticles used in this study had no significant effect on the growth of hMSCs, as reflected in both the K1 coefficient and the measurement of the final increase in cell number. Cell activity (K2) decreased significantly compared with the control group when cultivation was performed in the presence of a high-concentration P-GOs sample and increased when the concentration of the bP-GOs sample was low. Visually, a trend of increasing cell activity with increasing particle size can be observed.

In previous studies on the effects of other types of nanoparticles on MSCs, it has been repeatedly shown that these cells are able to internalize small particles (up to 200 nm) [38,39,40]. However, previous studies have predominantly used spherical polymer or metal particles. There are far fewer studies investigating the effect of graphene nanoparticles and their derivatives on MSCs. An important difference between graphene-based nanoparticles and those mentioned above is the two-dimensional structure of graphene-based materials, which provides a large surface area and the possibility of obtaining particles in different shapes.

The shape of the particles and surface modifications play an important role in achieving low cytotoxicity. For example, reduced GO nanoribbons were shown to cause more extensive DNA damage in hMSCs after a short incubation than nanosheets of similar size [41]. In another study using nanoparticles of PEG-coated reduced GO, incubation of cells for an extended period of time (up to 72 h) did not result in a significant decrease in viability [42]. In this case, particles with a diameter of 1 μm were used at concentrations of 5, 10, 50, and 100 μg/mL. The results obtained in our study are not consistent with these data. This could be due to several reasons, including the use of PEG with different lengths for nanoparticle coating. Syama et al. used PEG with a molecular weight of 1.9 kDa, whereas our particles were coated with PEG which had a molecular weight of 5 kDa.

The uptake of 2.7 μm-sized capsules by MSCs has previously been associated with a decrease in cell mobility [43]. The decrease in velocity was shown to be positively correlated with the concentration of capsules in the medium during incubation. At the same time, some studies have shown the ability of nanoparticles to accumulate in endosomes, leading to a change in cellular metabolism, stimulation of cells, and an increase in their mobility [44]. In this study, it was shown that incubation of cells with P-GOb and bP-GOb samples resulted in an increase in cell velocity, which may indicate that these particles have some effect on cellular metabolism, although the mechanism of this effect is not clear at present.

In general, there are good prospects for the use of graphene nanoparticles and their derivatives in diagnostics, therapy, and regenerative medicine. At the same time, the effect of these particles on human stem cells is still insufficiently studied. We see the need for further research in this area to gain a more comprehensive understanding of the parameters that play a key role in the cyto- and genotoxicity of graphene-based nanoparticles with respect to human stem cells. It is critical to know the consequences of short- or long-term exposure of human tissues to these materials.

## 4. Materials and Methods

**Obtaining a cell culture.** To study the effect of GO nanoparticles, a culture of human multipotent mesenchymal stem cells was obtained from a lipoaspirate at the Center for Immunology and Cellular Biotechnology at Immanuel Kant Baltic Federal University. The culture obtained met the minimum criteria set by the International Society for Cell Therapy [22]. The cells were stored in liquid nitrogen. After removal from the cryobank, the tube containing the frozen cell suspension was thawed in a water bath at 37 °C for 5 min. After thawing, the cells were washed with DMEM/F-12 containing 15 mM Hepes (Sigma-Aldrich, Saint Louis, MO, USA) by centrifugation at 1500 rpm for 5 min. Then, cells were transferred to T75 culture flasks (Eppendorf, Hamburg, Germany) and grown to 80% in complete culture medium (CCM) based on αMEM base medium (Sigma-Aldrich, Saint Louis, MO, USA) supplemented with 10% FBS (Sigma-Aldrich, Saint Louis, MO, USA), 100 U/mL penicillin, 100 μg/mL streptomycin (Thermo Fisher Scientific, Waltham, MA, USA), and 2 mM L-glutamine (Sigma-Aldrich, Saint Louis, MO, USA).

**Properties of graphene oxide nanoparticles.** Graphene oxide nanoparticles with sizes of 100–200 nm (GOs) and 1–5 μm (GOb) (“Ossila Ltd.”, Sheffield, UK) coated with linear (P) and branched (bP) polyethylene glycol (PEG) were used. The procedures for modification and characterization of nanoparticles have been previously described by the authors [45]. The following particles were used in the study: P-GOs (Ø 184 ± 73 nm), bP-GOs (Ø 287 ± 52 nm), P-GOb (Ø 569 ± 14 nm), and bP-GOb (Ø 1376 ± 48 nm). The information about the types of nanoparticles used in this study is summarized in Table 1. Further information is available in [45] and Appendix C.

**Cultivation of cells for viability assessment.** Cultivation was performed in CCM at 37 °C in a humidified atmosphere with 5% CO_2_. Cells were cultured in a 6-well plate with 5 mL CCM per well. Cell suspension was added to the wells to a final concentration of 1 ∗ 10^5^ cells per well. In addition, a suspension of GO nanoparticles of each type (P-GOs, bP-GOs, P-GOb, bP-GOb) was added to the wells at final concentrations of 5 and 25 μg/mL. Each group was present in duplicate. Cells without the addition of GO nanoparticles were used as negative controls. In addition, the cells with the addition of GO nanoparticles of each type at the maximum concentration (1 well for each sample type) were used as a null control.

**Evaluation of cell culture viability.** Analysis of cell viability before and after cultivation was performed by flow cytometry using MACS Quant FL7 (Miltenyi Biotec, Bergisch Gladbach, Germany). Guava ViaCount dye (Millipore, Burlington, MA, USA) was used for staining. This is a commercial reagent that combines an intercalating dye that selectively penetrates the membrane of dead cells and a specific dye for surface receptors, allowing accurate determination of the relative numbers of live, dead, and apoptotic cells.

Adherent culture of human mesenchymal stem cells was removed from the culture flasks by enzymatic treatment. The flask was washed 3 times with Hanks solution (HBSS) without magnesium and calcium (Capricorn Scientific, Ebsdorfergrund, Germany). Enzymatic treatment was performed with 3 mL of trypsin-EDTA solution (Sigma-Aldriche, Saint Louis, MO, USA). The gating strategy for flow cytometry is shown in Figure 6. Before the experiment, the viability of the cell culture was 90%. To compare the effect of the samples on apoptosis and cell death, the ratio between the number of dead cells and the number of apoptotic cells (D/A) was calculated.

**Evaluation of nanoparticle adhesion/internalization by cells.** In addition to viability, adhesion/internalization of particles by cells was also assessed. Cells that adhere/internalize GO nanoparticles became more granular, which increased the lateral scattering of light and could be detected by flow cytometry. To compare particle adhesion/internalization between samples, a subpopulation of high-granularity cells was isolated in the cell gate and the percentage of high-granularity cells was compared to that of the control group.

**Cultivation of cells for assessment of their behavioral changes in the Cell-IQ system.** Cultivation was performed in CCM at 37 °C in a humidified atmosphere with 5% CO_2_ for 24 h. Cells were cultured in 24-well plates with 1 mL CCM per well. A suspension of cells was added to the wells of the cell culture plate at a concentration of 2 ∗ 10^4^ cells per well. Then, a suspension of GO nanoparticles of each type was added to the wells to a final concentration of 5 and 25 μg/mL. Each group was present in triplicate. Cells cultured in CCM without addition of GO nanoparticles served as the control group.

For each well of the cell culture plate, four areas were selected for visualization (Figure 7). Images of these areas were taken every 30 min. The acquired images were analyzed to assess the activity of the cells. To obtain a better visual representation of the events taking place in each visualization area, the images for each of these areas were combined into a video file (one file per area, Appendix A). As a result of image processing, the dynamics of cell growth and migration activity of human mesenchymal stem cells was evaluated.

**Evaluation of the dynamics of cell growth and migration activity of cells in the Cell-IQ system.** Cells were counted in every second image acquired with the Cell-IQ system. Coefficients K1 and K2 were used to evaluate the dynamics of changes in cell number per area of visualization.

The K1 coefficient reflects changes in the number of cells in the observation area per unit time. It was calculated as the average value of the difference between the number of observed cells in each pair of analyzed images divided by the time interval between these images. In addition to K1, the final change in cell number in each visualization area was assessed. For this purpose, the ratio between the number of cells at the end of cultivation (after 24 h) and the number of cells at the beginning of cultivation was calculated for each area. This ratio reflects the total cell growth that occurred during cultivation.
K1 = Σ ((k^i^_start_ − k^i^_end_)/t^i^),(1)

k^i^_start—_number of cells observed in a single area of visualization at the beginning of a time period; 

k^i^_end_—number of cells observed in a single area of visualization at the end of a time period;

t^i^—time between observations.

The coefficient K2 reflects the activity of cell movement in the area of observation. It was calculated based on K1, but the absolute value of the difference between the number of cells observed in each pair of analyzed images was used. While K1 is influenced by cell growth, K2 is more appropriate to evaluate the migration of cells. If the average number of cells remains constant, their migration rate to/from the observation area can still be evaluated.
K2 = Σ (|k^i^_start_ − k^i^_end\_|/t^i^),(2)

k^i^_start_—number of cells observed in a single area of visualization at the beginning of a time period;

k^i^_end_—number of cells observed in a single area of visualization at the end of a time period;

t^i^—time between observations.

The speed of human mesenchymal stem cells was also evaluated. For this purpose, 3 cells were selected in each visualization area (if possible). Then, the central points of these cells in each image were determined. By tracking the changes in the coordinates of these points over time, the total distance travelled by each cell and its corresponding speed of movement could be calculated.

**Statistical analysis.** Descriptive statistics and hypothesis testing were performed using the standard STATISTICA package for Windows 10.0. Before data were analyzed, they were tested for normality using the Kolmogorov–Smirnov method. Median (Me), 25% (Q1), and 75% (Q3) quartiles were calculated for data that did not follow the normal distribution. Plots were generated with the R programming language [46] using the ggplot2 module [47].

For samples that did not follow the normal distribution, the significance of differences was assessed using the Wilcoxon T-test for pairs of dependent samples. Differences were considered significant when *p* < 0.05.

## 5. Conclusions

We studied the interaction of graphene oxide nanoparticles with hMSCs in the Cell-IQ in vivo monitoring system. We have shown that all the studied GO nanoparticles after 24 h of incubation have a cytotoxic effect on hMSCs at a high concentration (25 μg/mL), however, only bP-GOb particles were able to reduce cell viability at a low concentration (5 μg/mL).

We also showed that small particles (P-GOs) at a high concentration (25 µg/mL) reduced cell motility, while large-sized particles coated with branched PEG (bP-GOb) at a similar concentration increased this parameter. It was demonstrated that larger particles of P-GOb and bP-GOb (5 and 25 μg/mL) increased the speed of hMSCs movement.

However, the particles did not affect the changes in cell count during cultivation. In general, GO nanoparticles can reduce the viability of hMSCs, but increase the rate of cell movement. Thus, we have demonstrated for the first time the interaction of GO nanoparticles with hMSCs in the Cell-IQ in vivo monitoring system.

## Figures and Tables

**Figure 1 molecules-28-04148-f001:**
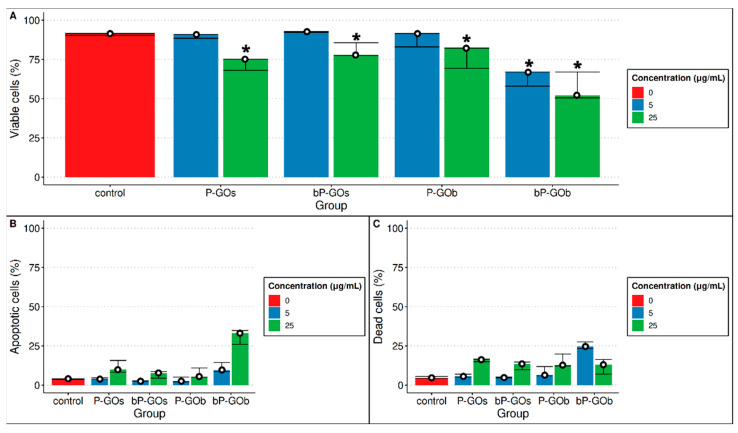
Comparison of the number of viable (**A**), apoptotic (**B**), and dead (**C**) cells after 24 h of cultivation in the presence of different types of GO nanoparticles (median, 1st and 3rd quartiles are shown). *n* = 2. Significant differences compared with control group (<0.05) are marked with *.

**Figure 2 molecules-28-04148-f002:**
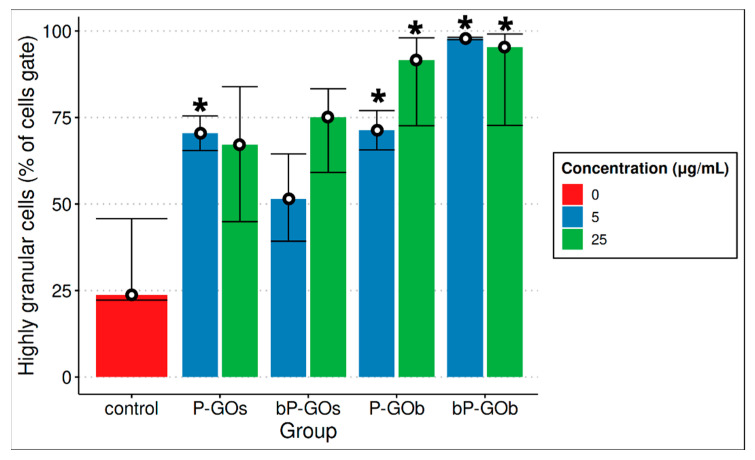
Comparison of the number of cells with high granularity after 24 h of co-cultivation with samples from GO (median is shown). *n* = 2. Significant differences compared with the control group (*p* < 0.05) are marked with *.

**Figure 3 molecules-28-04148-f003:**
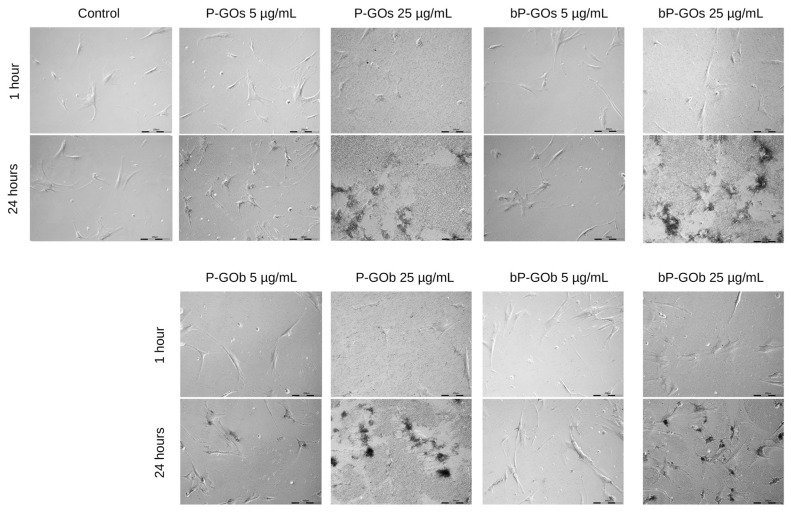
Photographs of cell culture plate wells taken in the Cell-IQ system. Each pair of images corresponds to the one area of visualization showing the changes during incubation of hMSCs with GO nanoparticles.

**Figure 4 molecules-28-04148-f004:**
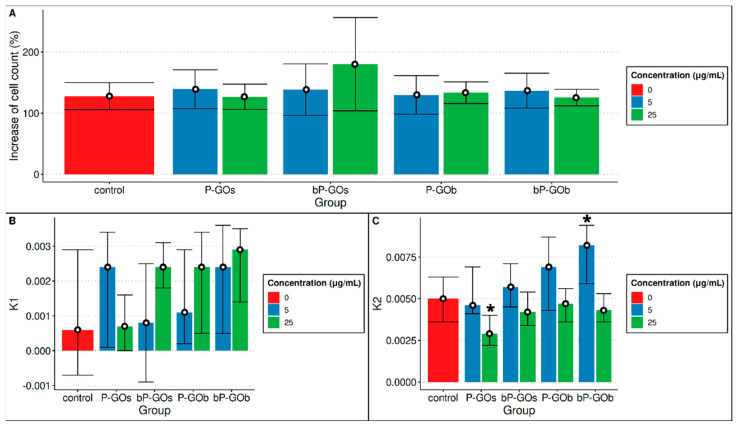
Comparison of (**A**) cell growth after 24 h of cultivation (mean and 95% confidence intervals shown); (**B**) change in cell number—K1 (median, 1st and 3rd quartiles shown). *n* = 12; (**C**) cell activity—K2 (median, 1st and 3rd quartiles shown). *n* = 12. Significant differences compared with control group (*p* < 0.05) are marked with *.

**Figure 5 molecules-28-04148-f005:**
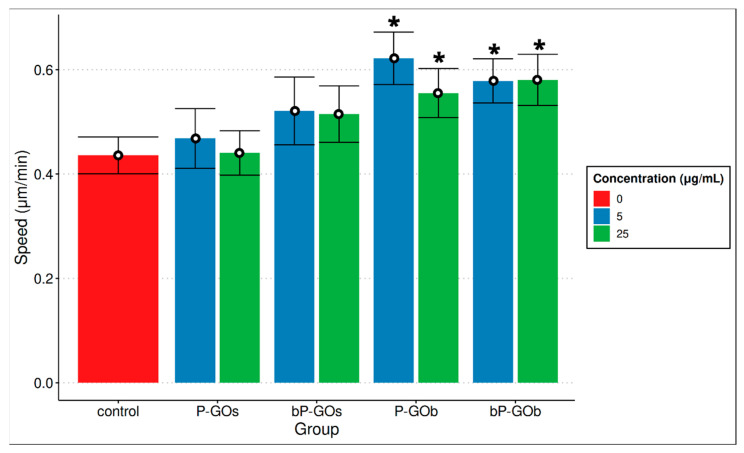
Comparison of the speed of cell movement during cultivation in the presence of different types of GO nanoparticles. Shown are the mean values and 95% confidence intervals of the mean values. *n* = 12. Significant differences compared with the control group (*p* < 0.05) are marked with *.

**Figure 6 molecules-28-04148-f006:**
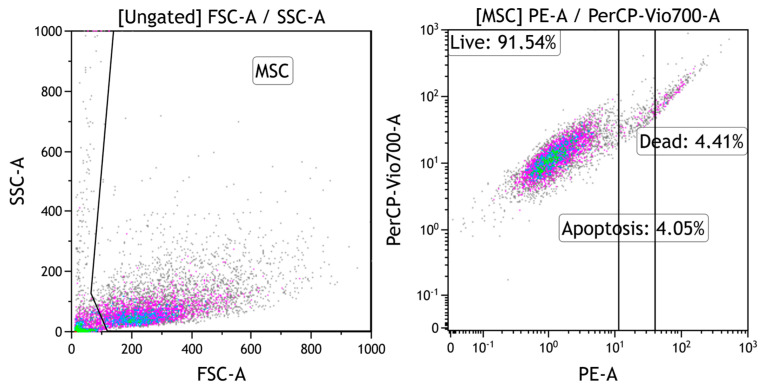
Gating strategy for human mesenchymal stem cells to assess the relative amounts of live, dead, and apoptotic cells.

**Figure 7 molecules-28-04148-f007:**
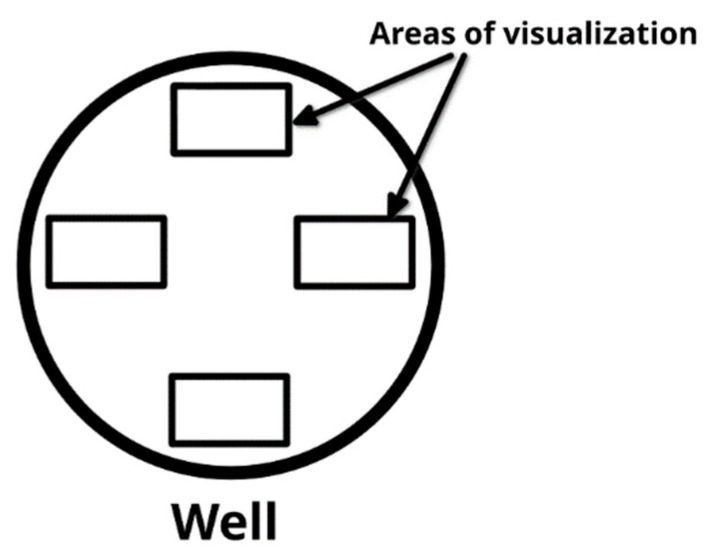
Distribution of visualization areas in a single well.

**Table 1 molecules-28-04148-t001:** Characteristics of GO nanoparticles.

Notation	Diameter (nm)	Coating
P-GOs	184 ± 73	Linear PEG
bP-GOs	287 ± 52	Branched PEG
P-GOb	569 ± 14	Linear PEG
bP-GOb	1376 ± 48	Branched PEG

## Data Availability

The datasets used and/or analyzed during the current study are available from the corresponding author on request.

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
