# Peer review of "Interaction of Graphene Oxide Nanoparticles with Human Mesenchymal Stem Cells Visualized in the Cell-IQ System"

_molecules, 2023, doi:10.3390/molecules28104148_

Round 1

Reviewer 1 Report

Potential toxicity of graphene oxide-based materials against hMSCs has been investigated. The subject is important and interesting. Therefore, it is suggested for publication. However, there are some points which should be clarified, addressed and/or discussed by the authors before the final publication. In this regard, I suggest revision of the manuscript based on the following points:

1.       The images shown in Figure 3 need scale bar.

2.       Figure 7 can be merged with other figures (such as Figure 3).

3.       The known mechanisms involved in the potential toxicity of nanomaterials especially graphene are: 1) physical direct interaction of extremely sharp edges of nanomaterials with cell wall membrane [Toxicity of graphene and graphene oxide nanowalls against bacteria], 2) ROS generation [RSC Adv., 2015,5, 80192-80195] even in dark [Langmuir 2015, 31, 33, 9155–9162], 3) trapping the cells within the aggregated sheets [Wrapping bacteria by graphene nanosheets for isolation from environment, reactivation by sonication, and inactivation by near-infrared irradiation] for bacteria and [Cyto and genotoxicities of graphene oxide and reduced graphene oxide sheets on spermatozoa] for spermatozoa, 4) oxidative stress [ACS Nano 2011, 5, 9, 6971–6980], 5) DNA damaging [Size-dependent genotoxicity of graphene nanoplatelets in human stem cells], and 6) contribution in generation/explosion of nanobubbles [Oxygen-rich graphene/ZnO2-Ag nanoframeworks with pH-switchable catalase/peroxidase activity as O2 nanobubble-self generator for bacterial inactivation]. These mechanisms should be addressed in the revised version. Then, the dominant mechanism occurred in this work should be discussed using suitable supports.

4.       No suitable evidence relating to the presence of graphene sheets and also their quality can be found in the manuscript. In this regard, Raman spectroscopy and analyzing the 2D band is highly necessary. For further help and support see [Carbon 81 (2015) 158-166] for GO and [Nano Lett. 2007, 7, 9, 2645–2649] for graphene sheets.

5.       Could the authors comment on the surface charge and also hydrophilicity of the samples? These parameters are important during the biological interactions of nanomaterials.

6.       It has been mentioned that “Graphene is a gapless semiconductor that is currently being actively used in microelectronics and materials science”. This can be further completed as follows: “Graphene is a gapless semiconductor that is currently being actively used in microelectronics, materials science and recently smartphones”. For supporting the last added term please see [Graphene as the ultra-transparent conductive layer in developing the nanotechnology-based flexible smart touchscreens] as a recent review in this regard.

7.       It has been mentioned that “GO can be used as a basis for nanoscale sensors that detect small molecules such as …”. This can be further completed by adding “DNA strains” based on [Toward single-DNA electrochemical biosensing by graphene nanowalls].

8.       It has been mentioned that “This makes GO a potential candidate for drug delivery and in vitro or ex vivo visualization that can be used to detect cancer”. This statement needs to be supported by, e.g., [Graphene nanomesh promises extremely efficient in vivo photothermal therapy]. Based on such works in the literature, the next statement, i.e., “However, the lack of controlled distribution capabilities and the inability to track particles in vivo complicate the use of GO as an effective drug delivery system.” seems as a mistake and should be removed and/or improved in the revised version. 

9.       It has been mentioned that “Human mesenchymal stem cells (hMSCs) … are undifferentiated cells with the ability to self-renew and differentiate into various mesenchymal tissues, mainly bone, cartilage, and adipose tissue.”. This needs to be supported as follows: “Human mesenchymal stem cells (hMSCs) … are undifferentiated cells with the ability to self-renew and differentiate into various mesenchymal tissues, mainly bone [Graphene nanogrids for selective and fast osteogenic differentiation of human mesenchymal stem cells], cartilage [Acta Biomaterialia Volume 96, 15 September 2019, Pages 271-280], adipose [DOI: 10.4172/2157-7552.1000212] and hepatic [Immobilization of modular peptides on graphene cocktail for differentiation of human mesenchymal stem cells to hepatic-like cells] tissues.”.

Some minor editions can be considered. 

Reviewer 2 Report

The authors in the paper evaluate the cytotoxic effects of GO nanoparticles using MSC in-vitro. The research is interesting however I doubt its applications until suitably justified. I have a few comments and would appreciate if the authors address them. Please cite references wherever possible.

1)      Is the aim of the paper to evaluate the cytotoxicity? Is there something special about the methodology? Can the authors highlight the novelty over other articles studying the toxicity of graphene oxide nanoparticles?

2)      Why were MSC chosen? Does using MSC reflect a general model for studying biocompatibility in-vitro?

3)      Can the authors explain the difference between M/A and D/A?

4)      Are the dead cells necrotic cells? What differentiates apoptotic and dead cells here?

5)      Can the authors provide a description as to how granularity indicates cell uptake? Can the authors use microscopy images to explain the process?

6)      Do the authors have data regarding the zetapotential of the nanoparticles?

7)      Nanoparticles suspended in the cell culture media will have different surface charge and protein corona composition, which should be different when nanoparticles are in the bloodstream in-vivo. Can the authors satisfactorily extrapolate the in-vitro observations reported here to in-vivo?

Please proofread and ensure that the manuscript does not have any grammatical errors.

Reviewer 3 Report

In this paper, the authors studied the interaction between graphite oxide (GO) nanoparticles and human mesenchymal stem cells (hMSCs) in the Cell IQ system, and evaluated the cell viability, mobility and growth rate. GO nanoparticles of different sizes coated with linear or branched polyethylene glycol (P or bP, respectively) were used at concentrations of 5 and 25 μg/mL. The authors found that all GO nanoparticles used in this study exhibited high concentrations (25 μ At g/mL, hMSCs exhibit cytotoxic effects, while at low concentrations (5 μ g/mL。 The authors also found that at 25 μ At a concentration of g/mL, P-GOs particles reduced cell mobility, while bP-GOb particles increased cell mobility. The larger particles (P-GOb and bP-GOb) increase the movement rate of hMSCs, regardless of concentration. There was no statistically significant difference in the growth rate of cells compared to the control group. I believe that publication of the manuscript may be considered only after the following issues have been resolved.

1.       The author needs to add the corresponding scale to the photo image in Figure 3.

2.       The format of the article needs to be greatly adjusted, and it needs to be adjusted according to the regular format of Molecules. For example, adjust the Materials and Methods section to the second part, and place Table A1-A2 in the attachment.

3.       What is Figure 7? We need to provide a ruler. Do you have a partially enlarged image?

4.       This paper focuses on the role of GO nanoparticles. Then, the author needs to provide relevant parameters of GO nanoparticles. Including morphology, crystal form, etc.

5.       The introduction can be improved. The articles related to some applications of graphene and graphene oxide materials should be added such as Results in Physics 48, 2023, 106420; Micromachines 2023, 14, 953; Diamond & Related Materials 128 (2022) 109273; Optics Express, 30(20), 35554-35566, 2022.

6.       Please check the grammar and spelling mistakes of the whole manuscript.

Minor editing of English language required

Round 2

Reviewer 1 Report

The revisions are acceptable. 

Some minor revisions can be considered. 

Reviewer 2 Report

I thank the authors for addressing my comments and queries.

Reviewer 3 Report

Accept in present form.